# IMU-Assisted Learning of Single-View Rolling Shutter Correction

**Jiawei Mo, Md Jahidul Islam, Junaed Sattar**
Department of Computer Science and Engineering
University of Minnesota, Twin Cities, United States of America
`[moxxx066, islam034, junaed]@umn.edu`

**Abstract:** Rolling shutter distortion is highly undesirable for photography and computer vision algorithms (*e.g.*, visual SLAM) because pixels can be potentially captured at different times and poses. In this paper, we propose a deep neural network to predict depth and row-wise pose from a single image for rolling shutter correction. Our contribution in this work is to incorporate inertial measurement unit (IMU) data into the pose refinement process, which, compared to the state-of-the-art, greatly enhances the pose prediction. The improved accuracy and robustness make it possible for numerous vision algorithms to use imagery captured by rolling shutter cameras and produce highly accurate results. We also extend a dataset to have real rolling shutter images, IMU data, depth maps, camera poses, and corresponding global shutter images for rolling shutter correction training. We demonstrate the efficacy of the proposed method by evaluating the performance of Direct Sparse Odometry (DSO) algorithm on rolling shutter imagery corrected using the proposed approach. Results show marked improvements of the DSO algorithm over using uncorrected imagery, validating the proposed approach.

**Keywords:** Rolling Shutter Correction, IMU, Learning

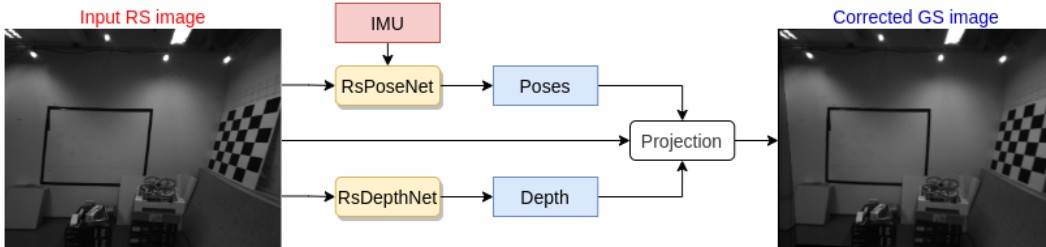

Figure 1: An overview of the proposed system. Given an RS image, the pixel-wise depth is generated by RsDepthNet; the row-wise poses are predicted by RsPoseNet using the RS image and IMU data. The pose estimates and depth maps are subsequently used for geometric projection to recover the corresponding GS image.

## 1 Introduction

Computer vision is one of the most commonly-used sensing modalities in robotics, empowered by the low cost, low power consumption of cameras, and the rich information they provide. For the underlying image readout mechanism, cameras either use a global shutter (**GS**) or rolling shutter (**RS**) system. The GS cameras capture the entire image at once, while the RS cameras capture the image row by row. As the camera can move arbitrarily when capturing the image for RS cameras, the pixels in different rows can be recorded at different camera poses, which leads to the RS distortions. Most vision algorithms are incapable of accounting for RS distortions and consequently perform poorly in their presence. However, most consumer-grade cameras (*e.g.*, smartphone cameras) are RS systems, which creates a challenge in making vision-based computing more ubiquitous. Even though

5th Conference on Robot Learning (CoRL 2021), London, UK.

some vision algorithms can be tuned for RS cameras, the adaptation process is non-trivial. For visual SLAM systems (*e.g.*, [1]), intermediate poses are usually interpolated for features in different image rows. RS distortions can thus be significantly detrimental to visual SLAM performance.

Alternatively, the RS images can be rectified before routing into the vision algorithms. Such RS correction algorithms can be categorized into multi-view methods [2, 3] and single-view methods [4, 5]. Traditionally, multi-view methods explore the relative geometry between images for RS correction; while the single-view methods rely on special features (*e.g.*, straight lines in [5]) due to the lack of geometric information. With the emergence of deep learning, several research works have addressed the problem of single-view automatic RS correction and have reported inspiring results [6, 7]. However, the accuracy and robustness of the existing methods for single-view RS correction are limited by the ill-posed nature of the problem. Besides, these approaches also adopt simplified formulations by making various assumptions on the camera motion such as restricting it in two-dimensions in [6] and assuming a constant velocity in [7]. Nevertheless, single-view RS correction is more appealing in real-world applications for its input simplicity. Moreover, the RS two-view geometry degenerates when the camera motion is pure translational [7], which is not uncommon in many robotic applications.

In this paper, we propose a novel deep neural network for correcting RS distortions; an overview of the system is illustrated in Fig. 1. In the proposed network, we estimate the depth of each pixel in an RS image and also recover its row-wise camera poses, which we subsequently use to reconstruct the corresponding GS image. This two-step process is analogous to the one by Zhuang *et al.* [7]. However, [7] incorporates a constant velocity assumption on camera motion which is not always valid in natural RS images (see Sec. 4.1); we propose to predict a camera pose for each image row to eliminate such assumption. To improve the accuracy and robustness of single-view RS correction, we propose to integrate the inertial measurement unit (IMU) data for pose prediction. An IMU estimates angular velocity and linear accelerations and is widely used in SLAM domain (*e.g.*, [8, 1]) as a complementary sensor to cameras. IMUs provide high-frequency motion cues that can help RS correction, and consequently, offers significantly more learning capacity for deep RS correction in more realistic scenarios. To train and test the proposed neural network, we generate a dataset from the TUM rolling shutter dataset [8], including RS images, IMU data, depth maps, row-wise camera poses, and corresponding GS images. Training the proposed neural network using this dataset, we show that RS distortions are correctly removed and the vision algorithm (*i.e.*, DSO [9]) works accurately on the resulting images.

In summary, the contributions of this work are the following: (*i*) a deep neural network that learns from single-view images and IMU data for accurate and robust RS correction; (*ii*) a novel dataset with real images and data for RS correction training; and (*iii*) extensive experimental validations of the proposed dataset and deep neural network for RS correction. Our implementation and dataset are available at `https://github.com/IRVLab/unrolling`.

## 2    Related Work

Removing RS distortions from images has several benefits as discussed in the previous section, and it is useful in computer vision, robotics, consumer mobile photography, etc.

Several neural networks have been proposed recently addressing the problem of single view RS correction. Rengarajan *et al.* [6] proposed a convolutional neural network architecture to predict the camera motion for RS correction. The predicted motions are up-sampled using a polynomial function to get row-wise components of camera motion, which is subsequently used to correct the RS distortion. This approach is feasible because the camera motion is limited to two dimensions (*i.e.*, $x$-axis translation and $z$-axis rotation). The mapping function relating the coordinates of the RS pixel to its undistorted correspondence is fully determined by this 2D motion. The work by Zhuang *et al.* [7] eliminates the 2D motion assumption above and enables full (6D) camera motion. They proposed the VelocityNet to predict the camera velocity, which is used to calculate the camera pose for each image row with the constant velocity assumption. Meanwhile, they adapt DispNet [10] (referred to as DepthNet) to predict pixel-wise depth in the RS image. With the camera pose and the depth, the undistorted GS image is generated by projection.

Other than single-view approaches, Liu *et al.* [3] proposed a network to remove RS distortion using two consecutive frames. They propose a motion estimation network to compute the cost volumes

between two consecutive frames, which are used to predict the pixel displacement between RS image and GS image. Karpenko *et al.* [11] proposed to use gyroscopes for video stabilization and RS correction based on classic geometry-based computer vision.

It is challenging to get a large set of GS/RS image pairs for RS correction training. Not only the GS/RS cameras need to have identical configures (*e.g.*, intrinsic parameters), but also they have to be co-located, which is mostly impossible in real world. Existing works use synthesized images instead. Rengarajan *et al.* [6] generate RS images from GS image datasets. For each GS image, they generate a random 2D camera motion and construct a mapping function to get the corresponding RS image. Subsequently, they train their neural network to perform the inverse action. Zhuang *et al.* [7], in the same vein, use a stereo GS image dataset to generate RS images. For each GS image pair, they run stereo matching to get a depth map; with a randomly-generated 6D camera velocity, they synthesize an RS image under the constant camera velocity assumption. Liu *et al.* [3] proposed two datasets for RS correction. One is from simulation; in the other dataset, the RS image is synthesized by sequentially copying a row of pixels from consecutive GS images captured by a high-speed (2400 FPS) camera, which is super expensive and its recording time is short.

The TUM rolling shutter dataset [8] has been proposed to benchmark visual-inertial SLAM with RS cameras, captured by the device shown in Fig. 2. The device is equipped with a pair of synchronized RS/GS cameras, an IMU, and a motion caption system recording ground-truth poses. However, the constant displacement between the RS camera and the GS camera prevents us from directly using the dataset for RS correction training, which we are going to solve in the following section.

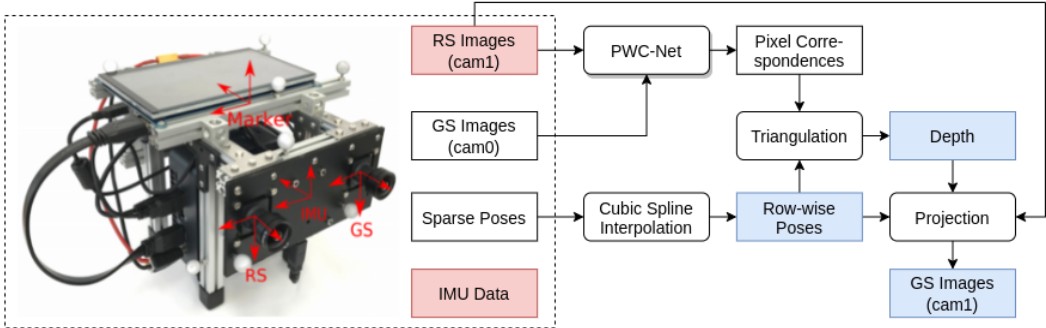

Figure 2: The data capture device used in [8] and an overview of the image processing pipeline. [Red: inputs for the neural network; Blue: ground truth for the neural network.]

## 3 Methodology

### 3.1 Dataset Generation

Currently, the TUM rolling shutter dataset [8] is the only publicly available dataset with synchronized RS/GS image pairs, camera poses, and IMU data. The RS/GS images are captured at 20Hz with $1280 \times 1024$ resolution; as shown in Fig. 2, we mark the GS camera as $\mathbf{cam_0}$ and the RS camera as $\mathbf{cam_1}$. The time difference between two consecutive rows in an RS image is reported as $29.4737\mu s$. The ground truth poses are recorded by a motion capture system running at 120Hz. The IMU runs at 200Hz. Please refer to [8] for more details about this dataset.

The RS images captured by $\mathbf{cam_1}$ and IMU data are used as inputs to our neural network. However, we cannot directly use the GS images from $\mathbf{cam_0}$ as ground truth because there is a constant displacement between the cameras. Hence, we recover the GS images at the location of $\mathbf{cam_1}$. For clarity, we mark the GS images captured from $\mathbf{cam_0}$ as $GS_0$ images and the recovered GS images as $GS_1$. As seen in Fig. 2, the idea is to recover a depth map from the stereo configuration and project points to the pose of the first image row of $\mathbf{cam_1}$ to get the $GS_1$ image.

Recovering depth from a $GS_0$ image and corresponding RS image is more challenging than from a pair of GS images. The rolling shutter distortion breaks the stereo epipolar constraint; even after stereo rectification, the pixel correspondences are not necessarily located in the same row, which makes conventional stereo matching algorithms (*e.g.*, [12]) inapplicable. Hence, we abandon the

stereo constraint and adopt PWC-Net [13], the state-of-the-art optical flow algorithm, to find the optical flow between the $GS_0$/RS image pairs. For robustness, we filter the correspondences by standard bi-directional matching.

In addition to the pixel correspondence, the relative camera poses are also required for depth recovery. This is because the rolling shutter mechanism makes the underlying camera poses no longer consistent for all pixels in the RS image. Unfortunately, the motion capture system is not efficient enough to capture a pose for every image row, even after we downsize the image resolution to $320 \times 256$ in this work. Since the motion capture system runs at 120Hz and it takes $1024 \times 29.4737\,\mu s = 30181.1\,\mu s$ to read out the entire RS image, we have about 3.6 poses for all 256 rows in the RS image, which is too sparse. To solve this problem, we adopt cubic spline interpolation [14] on the sparse poses to get a smooth pose for each row in the RS image. Since lens distortion is corrected during image preprocessing, we maintain a distortion lookup table to find the original scan-line when querying the pose for each pixel; that is, each transformation matrix ($\mathbf{T}$) in the following equations already considers the lens distortion lookup table implicitly.

With the pixel correspondence $(\mathbf{u}_{GS_0}, \mathbf{u}_{RS})$ from PWC-Net and the row-wise camera pose $\mathbf{T}^{GS_0}_{\mathbf{u}_{RS}}$ from cubic spline interpolation, we get the depth $d_{RS}$ and corresponding 3D point $\mathbf{X}_{RS}$ by solving the following triangulation problem:

$$d_{RS}\begin{bmatrix}\mathbf{u}_{RS} \\ 1\end{bmatrix} = \mathbf{K}_{RS}\mathbf{X}_{RS},$$

$$d_{GS_0}\begin{bmatrix}\mathbf{u}_{GS_0} \\ 1\end{bmatrix} = \mathbf{K}_{GS_0}\mathbf{T}^{GS_0}_{\mathbf{u}_{RS}}\begin{bmatrix}\mathbf{X}_{RS} \\ 1\end{bmatrix}.$$

Here, $\mathbf{K}_{RS}$ and $\mathbf{K}_{GS_0}$ are known camera intrinsic parameters. We also considered the depth filter [15] for further refinements; however, the improvement margins were insignificant, and hence it is omitted in our final implementation. Finally, we recover the $GS_1$ image ($\mathbf{u}_{GS_1}$) by projecting the 3D points $\mathbf{X}_{RS}$ to the pose of the first row in RS frame (*i.e.*, $\mathbf{T}^{GS_1}_{\mathbf{u}_{RS}} = \mathbf{T}^{RS_{row0}}_{\mathbf{u}_{RS}}$) as

$$d_{GS_1}\begin{bmatrix}\mathbf{u}_{GS_1} \\ 1\end{bmatrix} = \mathbf{K}_{RS}\mathbf{T}^{GS_1}_{\mathbf{u}_{RS}}\begin{bmatrix}\mathbf{X}_{RS} \\ 1\end{bmatrix}. \tag{1}$$

## 3.2 Network Architecture

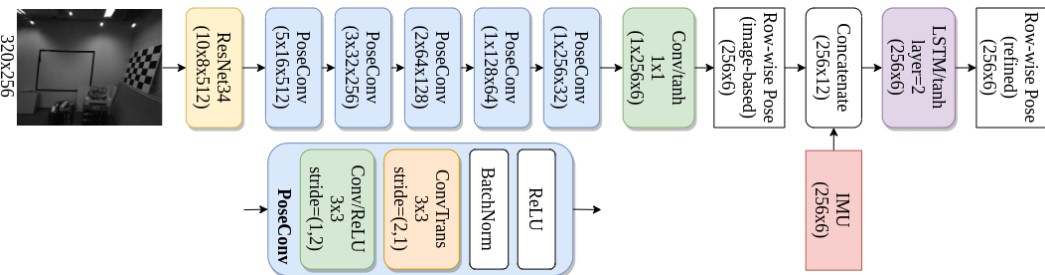

Figure 3: The architecture of RsPoseNet: a feature extraction network (ResNet-34) is followed by `PoseConv` blocks that learn row-wise poses, which is refined by IMU data using a LSTM network.

As mentioned earlier, our supervised training pipeline for learning rolling shutter correction is inspired by Zhuang *et al.* [7]. We outline the end-to-end pipeline in Fig. 1. Here, we adapt the DispNet [10] as our `RsDepthNet` (*i.e.*, DepthNet in [7]) for depth estimation. To predict row-wise poses (*i.e.*, the $\mathbf{T}^{GS_1}_{\mathbf{u}_{RS}}$ in Eq. 1), we propose a novel deep visual model named `RsPoseNet`; its detailed architecture is illustrated in Fig. 3. First, we use ResNet-34 [16] to extract high-level semantic features from a given input image. Then, we use a series of `PoseConv` layers to exploit those high-level features and learn row-wise poses. PoseConv consists of a convolution layer, a deconvolution (`ConvTrans`) layer, a batch normalization [17] layer, and a `ReLU` [18] activation layer. Using `stride` = $(1, 2)$ for the convolution layer, we combine information along each row; whereas using `stride` = $(2, 1)$ for the deconvolution layer, we expand the poses to infer a prediction for each row. Following that, we use a convolution layer and a `tanh` activation layer to finalize the poses. We use `tanh` as the final activation function as we normalize the poses to the range of $[-1, 1]$.

We will show empirically in Sec. 4.3 that predicting row-wise poses purely from images is not very accurate; hence, we propose to refine the poses by using IMU data. For unified input, we rotate the IMU data to the RS camera frame; we also interpolate the IMU data for each image row since IMU data is sparse running at a different frequency. The processed IMU data is then concatenated with the predicted poses from the image and fed into the 2-layer LSTM [19] for the final row-wise poses.

## 3.3 Training Details

Since our goal is to enable vision algorithms (*e.g.*, visual SLAM) on RS cameras, we adopt the end-point error (EPE) in [7], a standard geometric loss, for training and validation. The EPE measures the average Euclidean distance between the distorted and undistorted pixels. It is defined as

$$EPE(I) = \frac{1}{|I|} \sum_{\mathbf{u} \in I} \left|\left| \Pi(\mathbf{u}, d, \mathbf{T}') - \mathbf{u}_{GS_1}(\mathbf{u}) \right|\right|_2, \tag{2}$$

$$\Pi(\mathbf{u}, d, \mathbf{T}') = \mathbf{K}_{RS}\mathbf{T}' \begin{bmatrix} d\mathbf{K}_{RS}^{-1} \begin{bmatrix} \mathbf{u} \\ 1 \end{bmatrix} \\ 1 \end{bmatrix}. \tag{3}$$

Here, for each pixel $\mathbf{u}$ in the input RS image $I$, we project it to the corresponding GS frame in Eq. 3, which is analogous to Eq. 1 but using predicted pose $\mathbf{T}'$. We compare the projection with ground truth $\mathbf{u}_{GS_1}(\mathbf{u})$ to get the EPE. Note that the $d$ here is the ground-truth depth, which decouples the training of RsDepthNet and RsPoseNet.

We initialize the ResNet-34 in RsPoseNet with the pre-trained weights from ImageNet classification [20]. We empirically find that training directly using EPE does not yield optimal results, hence we initialize RsPoseNet using pose loss (mean squared error of the row-wise poses) for 50 epochs with a learning rate of $1e-3$; then we refine RsPoseNet using the EPE for another 50 epochs with $1e-4$ learning rate. Pose loss is straightforward for training, but a minimized pose loss does not guarantee a minimized EPE because rotation and translation play different roles in projection. Refining RsPoseNet with EPE optimizes the weights in each dimension of poses to minimize the EPE, which is meaningful and necessary.

# 4 Experimental Evaluation

By applying the data generation process described in Sec. 3.1 on the TUM rolling shutter dataset [8], we get 10 sequences of RS images with corresponding IMU data, depth maps, row-wise camera poses, and recovered $GS_1$ images. There are 9495 frames in total, which is slightly less than the total number of frames (9523) in the TUM rolling shutter dataset, because some frames at the beginning or end of a few sequences were not recorded by the motion capture system, hence they were discarded.

## 4.1 Data Verification

In the evaluation, we quantify RS distortion by the EPE formulations defined in Eq. 2 and Eq. 3. Since EPE is evaluated with respect to the $GS_1$ images generated by the proposed method, we need to verify the validity of $GS_1$ images. To do so, we run DSO [9] on a continuous sequence of images; being a direct method for visual odometry, DSO is very sensitive to RS distortion. Fig. 4 shows the qualitative results for a particular sequence (#02) in the TUM dataset; as the top-left block (RS) illustrates, DSO completely fails on the RS images with an EPE of **5.738** pixels. The scene in the bottom-left block ($GS_1$ gt) is reconstructed by DSO on our recovered $GS_1$ images, which clearly shows the accurate room layout; EPE for this case is zero by definition. Moreover, we run DSO 10 times and calculate the average absolute trajectory error [21] (ATE) with $Sim(3)$ alignment. The ATE on the RS images is **0.269** meters, while it reduces to **0.047** meters on the $GS_1$ images. These qualitative and quantitative results validate our data generation process and justify the use of EPE as an evaluation metric. We also generate data with the constant velocity assumption as in [7]; here we verify that its EPE is **0.228** pixels. The reconstructed scene in the top-right ($GS_1$ gt_cv) of Fig. 4 is not as good as the one without such assumption, which is further validated by the fact that its ATE=**0.157** meters. This shows that a constant velocity assumption is sub-optimal in real-world settings. Table 1 quantitatively summarizes these experimental results.

Table 1: EPEs and ATEs of sequence 02 on RS images, $GS_1$ images, $GS_1$ images with constant velocity assumption, and $GS_1$ images estimated by our network.

| Images | RS | $GS_1$ | $GS_1$ gt_cv | $GS_1$ pred |
|---|---|---|---|---|
| EPE (pixels) | 5.738 | 0 | 0.228 | 0.937 |
| ATE (meters) | 0.269 | 0.047 | 0.157 | 0.064 |

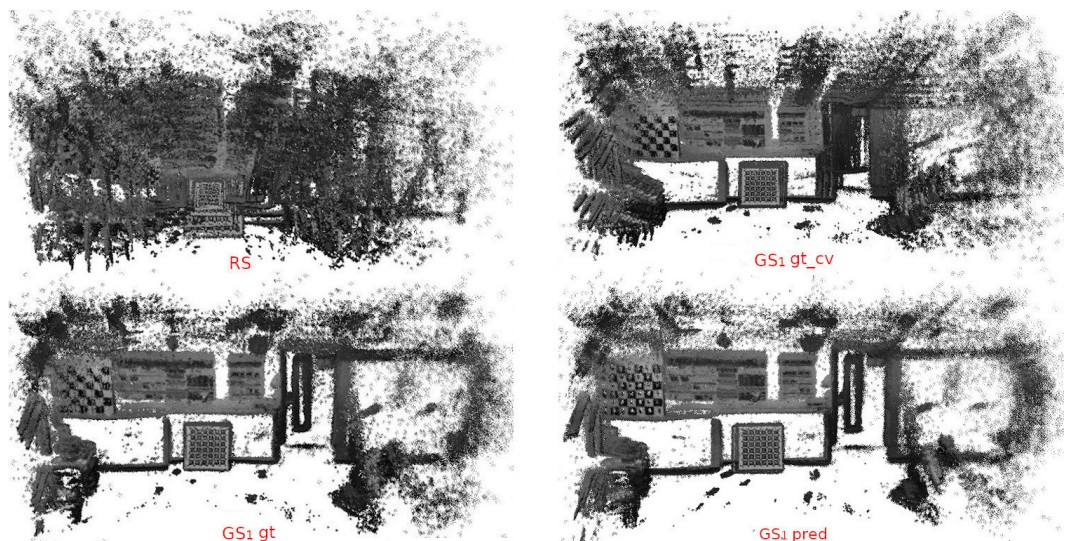

Figure 4: Top-to-bottom, left-to-right: Scenes reconstructed by DSO on RS images (RS), $GS_1$ images ($GS_1$ gt), $GS_1$ images with constant velocity assumption ($GS_1$ gt_cv), and $GS_1$ images corrected by our network ($GS_1$ pred).

## 4.2 Network Evaluation

We split the dataset as follows for training and testing: we use sequence 02 and sequence 07 for testing; the remaining 8 sequences are used for training and validation, among them, we reserve the middle 10% data for validation and the rest for training. Consequently, we have 7362 frames for training, 815 frames for validation, and $534 + 784 = 1318$ frames for testing. Our network implementation is based on Keras [22] and trained on a Nvidia™ Titan V GPU with 12GB memory. We mainly focus on [7] (denoted as SMARSC) for comparison with the goal of showing the contribution of IMU data for RS correction. The SMARSC is also trained with the EPE. Additionally, we fine-tune the state-of-the-art two-view learning-based RS correction approach [3] (denoted as TwoView) on our dataset for comparison. However, this approach outputs the corrected GS image directly so that we cannot accurately calculate EPE; thus we include it for qualitative comparison.

Table 2: The ratio of images whose RS distortion is reduced and EPE (in pixels) on the test data.

(a) SMARSC [7] vs. Ours

| Seq. | | Input | SMARSC [7] | Ours |
|---|---|---|---|---|
| 02 | EPE | 5.738 | 2.599 | **0.937** |
| | Ratio | - | 90.4% | **99.3%** |
| 07 | EPE | 6.593 | 2.267 | **0.746** |
| | Ratio | - | 86.5% | **95.9%** |

(b) Tests on partial IMU data.

| None | Acc. | Gyr. | Full |
|---|---|---|---|
| 2.327 | 2.939 | 0.977 | **0.937** |
| 93.1% | 90.4% | **100%** | 99.3% |
| 1.802 | 1.958 | 0.896 | **0.746** |
| 85.6% | 86.1% | **96.0%** | 95.9% |

Table 2a reports the performance comparisons on our test data. The term *Ratio* represents the percentage of testing images whose RS distortion (measured by EPE) gets reduced. For SMARSC [7], the EPE of sequence 02 is reduced by half and it predicts RS distortion in the correct direction for more than 85% of the tests. It shows that learning RS correction from single-view images is feasible. As for the proposed method (*i.e.*, 'Ours'), it predicts RS distortion in the correct direction for more

than 95% of the tests with sub-pixel EPE, which is a significant improvement over SMARSC [7]. As we will discuss in Sec. 4.3, the performance improvement mainly comes from the IMU data. We present some qualitative results of RS correction by the proposed system in Fig. 5a. The predicted GS images look very similar to the ground truth images, as most of the visible RS distortions are removed. For TwoView [3], even though it predicts the correct direction of RS distortions, the corrected results introduce new undesired image distortions. Fig. 5b confirms that the new distortion comes from the TwoView [3] itself instead of the process of fine-tuning on our dataset.

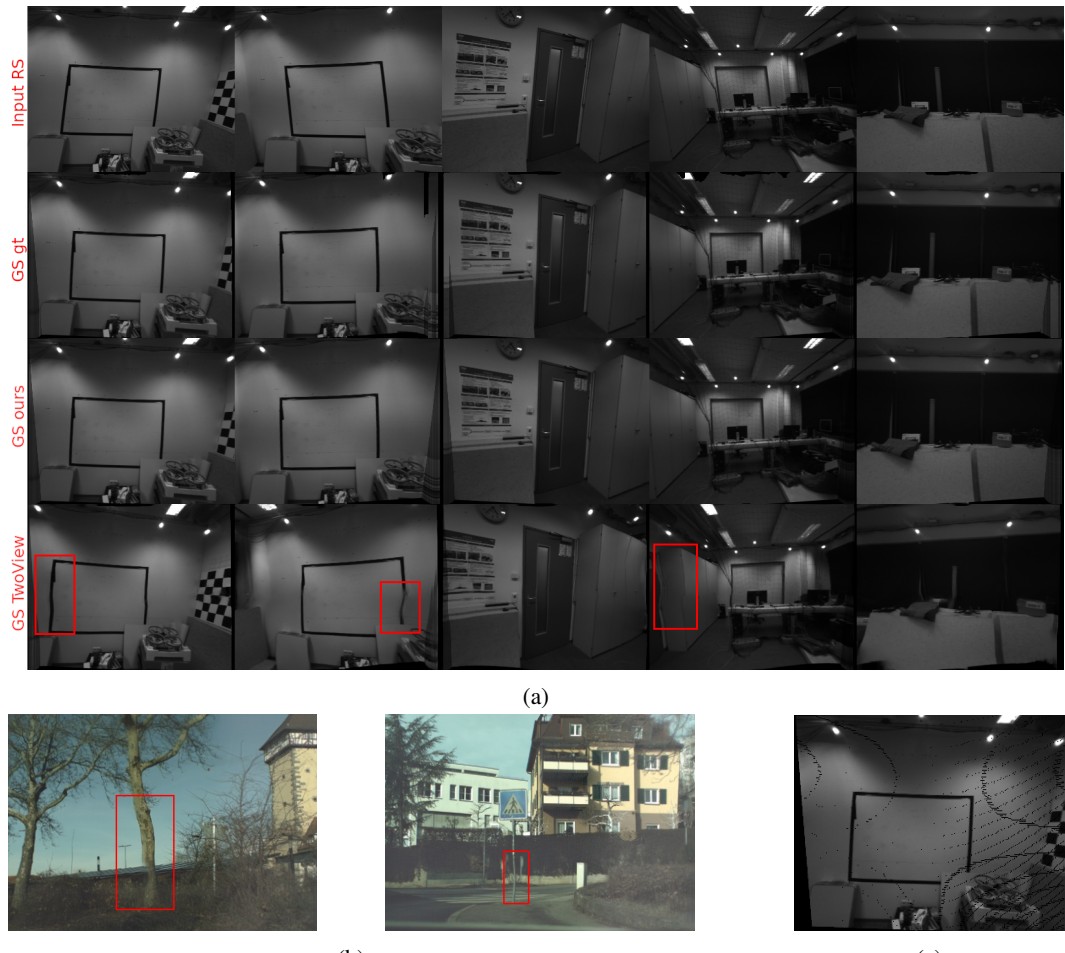

Figure 5: Qualitative Results. (a): A few samples of predicted GS images. From top to bottom are: input RS images, the ground truth $GS_1$ images, the GS images corrected by our network, and the GS images corrected by TwoView [3]. The last row contains new undesired distortion. (b): The undesired distortion also appears in off-the-shelf results [3]. (c): Rectified GS image by projection without post-processing. The projected image is incomplete with many missing pixels (black dots).

**Application Study** We evaluate the performance of DSO on GS images recovered by our method. The reconstructed scene is shown in the bottom-right of Fig. 4; it is as good as the one on ground-truth $GS_1$ images and significantly better than the one on the original RS images. We get the average ATE of **0.064** meters, which is much smaller than the ATE on RS images (**0.269** meters) and very close to the ATE on ground-truth $GS_1$ images (**0.047** meters). From the quantitative and qualitative results, we show that the proposed RS correction network enables downstream vision algorithms (DSO in this case) to run on RS cameras without any modification. We also run DSO on the results by SMARSC [7] and by TwoView [3] but DSO fails consistently. For computational efficiency, the network inference is able to run in real-time ($\sim$ 60 FPS on our Titan V GPU), including the projection of pixel from the input RS image to the rectified GS image (*i.e.*, Eq. 3). However, since the projection is not surjective [23] (not all GS pixel is guaranteed to be filled), the output GS image

is incomplete as shown in Fig. 5c. We apply interpolation to complete the output image, with Fig. 5a showing the results after interpolation. As interpolation is computationally expensive ($\sim$ 20 FPS on CPU), we will explore alternatives (*e.g.*, median filter) for higher computational efficiency in future work.

### 4.3 Ablation Study on IMU Data

To analyze the contribution of IMU data for RS correction, we disable the gyroscope, or the accelerator, or both when feeding the IMU data into the network; we re-train the network and report the results in Table 2b. Using gyroscope alone (*i.e.*, 'Gyr.') already yields good results, while integrating the accelerator data (*i.e.*, 'Full') further reduces the EPE slightly. However, the accuracy of using accelerator data alone (*i.e.*, 'Acc.') is low. Gyroscope measures the camera motion more directly than accelerator does (angular velocity versus linear acceleration), thus gyroscope being more important for RS correction is expected. Nevertheless, even when the IMU data is completely disabled (*i.e.*, 'None'), it still slightly outperforms SMARSC [7]. The potential reason is that the row-wise poses are highly correlated with the constant velocity assumption so that the velocity error affects every pixel, whereas row-wise poses are predicted individually in our method.

### 4.4 Generalization Performance

We also test our network on the WHU RS Visual-Inertial Dataset [24], which contains RS/GS image pairs, IMU data, and ground-truth poses. We use the training results on our proposed dataset without fine-tuning on the WHU dataset to test the generalization performance of our RS correction network. Fig. 6 gives some qualitative results. One important observation is that when the camera rotates, our network works most of the time, with the predicted GS images well-aligned to the ground-truth GS images as illustrated in the left part of the figure; however, for pure translational motion, our method is more likely to fail (*e.g.*, right part of Fig. 6). The potential reason is that pure translational motion is very rare in our training dataset from the TUM RS dataset. For future work, we plan to increase generalization capability by augmenting the training set with more diverse data.

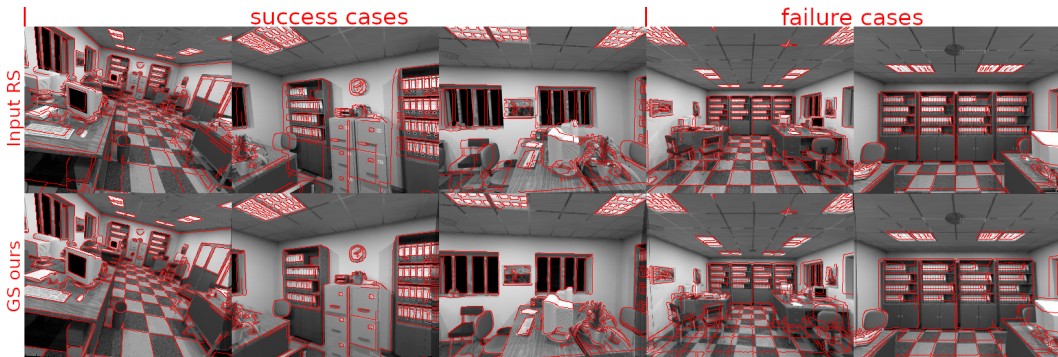

Figure 6: Qualitative results of our network on WHU RS Dataset [24]. The top row shows the input RS images; the bottom row shows the GS images corrected by our network. The edges of ground-truth GS images (red lines) are overlaid on the results for RS comparison.

## 5 Conclusions

We propose a novel neural network for single-view rolling shutter correction, where we use IMU data to significantly improve rolling shutter pose prediction. Rather than synthesizing rolling shutter images, we generate real images to train the proposed network. Our experiments validate the improved accuracy and robustness in rolling shutter correction and downstream vision algorithms by using the proposed approach. For future work, we will extend the proposed method for diverse environments by designing our own data acquisition device. We also intend to deploy the proposed approach for vision-based robot localization and odometry purposes while using inexpensive and ubiquitous rolling shutter cameras.

**Acknowledgments**

This work was supported by the US National Science Foundation Award IIS #1637875, the University of Minnesota Doctoral Dissertation Fellowship, and the MnRI Seed Grant.

We thank Igor Slinko for sharing the weights of PWC-Net on grayscale images with us. We are also thankful to Nvidia[TM] for their donation of GPUs to support our work.

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
