# OpenReview forum: "IMU-Assisted Learning of Single-View Rolling Shutter Correction"
_robot-learning.org/CoRL/2021/Conference — CoRL2021 Poster_

### Official Review · Reviewer_DMMe · 2021-07-23

**Originality:** Good
**Technical Quality:** Very Good
**Clarity Of Presentation:** Very Good
**Impact:** 3

**Recommendation:**

Weak Accept: I recommend accepting the paper, but will not argue for my recommendation if the majority of other reviewers have a different opinion.

**Summary:**

This paper proposes an imu-assisted, learning-based rolling shutter correction method. The key to the approach is two neural networks - one that predicts a depth map for the rolling shutter image (RsDepthNet) and one that predicts a per-row pose (RsPoseNet). The output of these networks is then used to reproject the rolling shutter pixels into an estimate of the global shutter image. The experimental results show that the method performs very well.

**Issues:**

I would like the authors to comment on the sensitivity of the approach to the depth prediction as this seems to be the main weakness of the approach.

**Reviewer Expertise:**

Good: General knowledge of the area

**Strengths And Weaknesses:**

**Strengths**
- The paper is clear and very well written. The figures are very well made and help the reader to easily understand the proposed method. The paper contains sufficient details about the implementation and parameters used.
- Unlike existing methods, the proposed approach makes very little assumptions on the motion of the camera. This is due to the fact that row-wise poses and pixel-wise depths are predicted with the help from neural networks. Therefore the method hypothetically will outperform most existing approaches which assume purely rotational, or other limited degrees of motion.
- The experimental evaluation is very thorough. The approach has been compared to the state of the art (SMARSC), and a nice ablation study has been done to investigate the effect that the IMU has on the accuracy of the approach.
- The results show that the method outperforms the state of the art (even when the IMU is not used, which is a fair comparison). The reasons for this are nicely explained and confirm the claims made in the paper.

**Weaknesses**
- The approach is very similar to [7]. The only differences are that the prediction is now conditioned on an IMU and per-row poses are predicted. This detracts (slightly) from the novelty of the paper.
- The generalization performance seems a bit poor. This is somewhat expected as the method relies on accurately predicting the depth of the rolling shutter image. Perhaps an evaluation of the sensitivity of the method to depth / pose prediction errors will help the reader to understand this aspect better.


**Summary Of Recommendation:**

Overall, this is a very useful method that seems to work well. While there is still some room for improvement in terms of generalization, this could possibly be addressed with better training data.

---

> ### Author Response · Authors · 2021-08-25
> **Thank you for reviewing our paper**
>
> Thank you very much for reviewing our paper. We appreciate your positive feedback, constructive comments, and important suggestions toward improving the paper. We have made the revisions accordingly; the comments below reflect our responses to the meta comments by the Area Chair, with some additional details.
>
> ---
>
> - Please note that our framework incorporates a novel design of the proposed model RsPoseNet to enable row-wise pose prediction and IMU integration toward rolling-shutter correction. Although the underlying problem formulation is inspired by [7], our model outperforms its baseline performance by considerable margins. More importantly, it provides significant performance improvement with IMU integration - which was the focus of our contribution. Thus, we believe that our work will have significant impact. To better reflect the impact, we modified Table 2 in the paper with input error for direct comparison; we also run DSO on the images corrected by [7] but it fails consistently (page 7, line 226).
>
> - While the generalization performance of our proposed framework falls short in few challenging cases, it can be improved by augmenting more diverse training samples because the proposed network successfully demonstrates the learning capacity for the underlying function. We certainly plan to address this in our future work, which we explicitly mention at the end of the experimental evaluations (Section 4) in the paper.
>
> - Your comments on sensitivity to depth/pose prediction is greatly appreciated. We actually conducted the experiments but the results were not included in the paper due to page limitations. We have provided the sensitivity results and discussions in our supplementary material as a standalone single-page PDF document. As can be seen in the results provided in that document, in the current experimental settings, the proposed approach is actually not very sensitive to depth prediction.
>
> ---
>
> We thank you again for your constructive feedback.

---

### Official Review · Reviewer_kkCY · 2021-07-24

**Originality:** Very Good
**Technical Quality:** Very Good
**Clarity Of Presentation:** Very Good
**Impact:** 2

**Recommendation:**

Weak Accept: I recommend accepting the paper, but will not argue for my recommendation if the majority of other reviewers have a different opinion.

**Summary:**

This paper is addressing the challenges of using global shutter cameras in slam. The problem with rolling shutter(RS) cameras are that each row is taken at a different time step and there will be a distortion when the camera moves and because of that it breaks the direct sparse odometry method. The paper argues that linear interpolation of IMU poses to estimate the position of the camera for each row of the image is not enough. They predict the depth for each row of the image and then fuse it with IMU data and further process it to predict per row pose. With the corrected pose they show the quality improves quite significantly. The model is trained on TUM rolling shutter dataset where there is a global shutter and rolling shutter cameras. Correspondences between RS and GS camera is found with optical flow (using PWC-Net) and then the pose is optimized to minimize the reprojection error between RS and GS after reprojection.

**Issues:**

N/A

**Reviewer Expertise:**

Fair: Some knowledge of the area

**Strengths And Weaknesses:**

Strengths:
- The paper tackles a practical problem for SLAM system
- The paper is well written and the method makes sense.
- The method makes significant improvement quantitatively and qualitatively.
- Comparison on different datasets and with different methods.
- I also like the ablation studies to show the effect of using IMU vs not using IMU.

Weaknesses:
N/A


**Summary Of Recommendation:**

I am not quite familiar with the literature in this area so I don’t know about potential missing comparisons. The improvement seems significant to me and the method makes sense. I'll let other expert reviewers to point out potential flaws.

---

> ### Author Response · Authors · 2021-08-25
> **Thank you for reviewing our paper**
>
> Thank you very much for reviewing the paper. We gratefully acknowledge and appreciate your positive feedback.

---

### Official Review · Reviewer_BmJn · 2021-07-26

**Originality:** Good
**Technical Quality:** Very Good
**Clarity Of Presentation:** Very Good
**Impact:** 3

**Recommendation:**

Weak Accept: I recommend accepting the paper, but will not argue for my recommendation if the majority of other reviewers have a different opinion.

**Summary:**

This paper performs rolling-shutter (RS) correction by estimating pixelwise depth and row-wise camera transform given a single rolling-shutter image, and this allows correcting the image to a global-shutter(GS) one. The key contributions include: a) using additional IMU information for more accurate prediction, b) leveraging a real dataset with motion information and RS-GS pairs for training. The empirical and qualitative results clearly demonstrate the benefits of the IMU sensor inputs and the system yields improvements over prior related work [7].

**Issues:**

A presentation concern is that the abstract claims that the paper 'releases' a dataset, but I think this is not an accurate claim. More precisely, the paper leverages an existing dataset with off-the-shelf systems to extract additional information e.g. 'ground-truth' depth.

**Reviewer Expertise:**

Fair: Some knowledge of the area

**Strengths And Weaknesses:**

**Strengths**

- The paper tackles a very relevant and basic task that can help improve any visual odometry approach. Towards this, it demonstrates clear benefits of including IMU sensor readings when inferring row-wise camera poses and this is an interesting finding which could be applicable to several methods.

- The overall approach is very sensible and intuitive. In particular, the main contributions of: a) extracting pseudo-GT labels for depth training using optical flow between RS and GS image, and b) modifying the pose prediction network to integrate IMU information, both make sense are well-executed.

- The ablations presented by the paper are convincing regarding the importance of the IMU information when predicting camera motion.

- The qualitative results also demonstrate a clear use-case for the proposed approach in allowing better registration by visual odometry approaches.

**Weaknesses**

- While the empirical results and the overall system developed are encouraging, the technical contributions of the work are limited to allowing an additional input (IMU readings) in existing frameworks e.g. [7].

- The presented generalization results (Figure 6) are not very impressive and I am unsure if the learned models have perhaps overfitted to the statistics of the training dataset.

- While using the E2E error is an intuitive metric, training by optimizing this metric makes the comparison slightly unfair to the baselines as this method is directly trained to optimize this metric.



**Summary Of Recommendation:**

Overall, I feel this paper is a well-motivated and well-executed one, and while the specific technical innovations maybe limited, the overall approach for extracting pseudo-GT for training and incorporating IMU readings for better camera prediction is a novel and well-ablated one. I think this therefore maybe a value contribution that could benefit the SLAM-based approaches.

---

> ### Author Response · Authors · 2021-08-25
> **Thank you for reviewing our paper**
>
> Thank you very much for reviewing our paper. We appreciate your positive feedback, constructive comments, and important suggestions toward improving the paper. We have made the revisions accordingly; the comments below reflect our responses to the meta comments by the Area Chair, with some additional details.
>
> ---
>
> - Please note that our framework incorporates a novel design of the proposed model RsPoseNet to enable row-wise pose prediction and IMU integration toward rolling-shutter correction. Although the underlying problem formulation is inspired by [7], our model outperforms its baseline performance by considerable margins. More importantly, it provides significant performance improvement with IMU integration - which was the focus of our contribution. Thus, we believe that our work will have significant impact. To better reflect the impact, we modified Table 2 in the paper with input error for direct comparison; we also run DSO on the images corrected by [7] but it fails consistently (page 7, line 226).
>
> - While the generalization performance of our proposed framework falls short in few challenging cases, it can be improved by augmenting more diverse training samples because the proposed network successfully demonstrates the learning capacity for the underlying function. We certainly plan to address this in our future work, which we explicitly mention at the end of the experimental evaluations (Section 4) in the paper.
>
> - Moreover, please note that the baseline approach [7] is also trained with the EPE metric (same as ours); hence, we believe that the evaluation comparison is reasonable. We modified the paper to explicitly mention this fact (page 5, line 160 and page 6, line 204).
>
> - Lastly, we have changed the ‘release’ word to ‘extend’ in the abstract. Thank you for pointing this out.
>
> ---
>
> We thank you again for your constructive feedback.

---

> > ### Comment · Reviewer_BmJn · 2021-09-03
> > **Thanks for the Response**
> >
> > I'd like to thank the authors for the response, and in particular the clarification regarding the baseline also being trained with the E2E metric. While generalization is an obvious concern that prevents me from more strongly arguing acceptance, I think the paper is overall interesting and well-executed, and would like to persist with the current weak accept rating.

---

### Meta-Review · Area_Chair_dSUp · 2021-08-11

**Recommendation:** Accept (Poster)
**Confidence:** 5

**Metareview:**

The paper studies an important problem since rolling-shutter correction can enhance the performance of visual-odometry and SLAM pipelines. The reviewers also appreciated the technical approach (e.g., integration of IMU information, pseudo-labels for depth), the clarity of the presentation, and the experimental results and ablation. The main weakness of the paper is that the experiments do not make a strong case for the generalization capability of the proposed approach. However, after the rebuttal, the authors added more discussion about generalization, which the reviewers and area chair found satisfactory.

---

> ### Author Response · Authors · 2021-08-25
> **Thank you for organizing and summarizing the reviews**
>
> Thank you very much for your kind remarks on our technical approach, presentation, and experimental analyses in the paper. We appreciate your time in organizing the reviews and summarizing the key points identified by the reviewers.
>
> ---
>
> - While the generalization performance of our proposed framework falls short in few challenging cases, it can be improved by augmenting more diverse training samples because the proposed network successfully demonstrates the learning capacity for the underlying function. We certainly plan to address this in our future work, which we explicitly mention at the end of the experimental evaluations (Section 4) in the paper.
>
> - Moreover, please note that the baseline approach [7] is also trained with the EPE metric (same as ours); hence, we believe that the evaluation comparison is reasonable. We modified the paper to explicitly mention this fact (page 5, line 160 and page 6, line 204).
>
> - Please note that our framework incorporates a novel design of the proposed model RsPoseNet to enable row-wise pose prediction and IMU integration toward rolling-shutter correction. Although the underlying problem formulation is inspired by [7], our model outperforms its baseline performance by considerable margins. More importantly, it provides significant performance improvement with IMU integration - which was the focus of our contribution. Thus, we believe that our work will have significant impact. To better reflect the impact, we modified Table 2 in the paper with input error for direct comparison; we also run DSO on the images corrected by [7] but it fails consistently (page 7, line 226).
>
> ---
>
> We thank you again for coordinating the reviews for this paper. We have responded to the respective reviewers' comments individually; please find the inline responses on the following pages.

---

### Decision · Program_Chairs · 2021-09-13

**Decision:**

Accept (Poster)

**Comment:**

The paper studies an important problem since rolling-shutter correction can enhance the performance of visual-odometry and SLAM pipelines. The reviewers also appreciated the technical approach (e.g., integration of IMU information, pseudo-labels for depth), the clarity of the presentation, and the experimental results and ablation. The main weakness of the paper is that the experiments do not make a strong case for the generalization capability of the proposed approach. However, after the rebuttal, the authors added more discussion about generalization, which the reviewers and area chair found satisfactory.